# Equity for health delivery: Opportunity costs and benefits among Community Health Workers in Rwanda

Janna M. Schurer[1,2]*, Kelly Fowler[2], Ellen Rafferty[3], Ornella Masimbi[4], Jean Muhire[5], Olivia Rozanski[2], Hellen J. Amuguni[1,2]

1 Center for One Health, University of Global Health Equity, Kigali, Rwanda, 2 Department of Infectious Disease and Global Health, Cummings School of Veterinary Medicine, Tufts University, North Grafton, MA, United States of America, 3 Faculty of Nursing, University of Alberta, Edmonton, Alberta, Canada, 4 Basic Medical Sciences, University of Global Health Equity, Kigali, Rwanda, 5 Environmental Health Sciences, University of Rwanda, Kigali, Rwanda

* jschurer@gmail.com

**Data Availability Statement:** All relevant data are within the paper.

**Funding:** This research was generously funded by a Cummings Foundation grant (V710458) awarded

## Abstract

Community Health Workers (CHWs) play a vital role delivering health services to vulnerable populations in low resource settings. In Rwanda, CHWs provide village-level care focused on maternal/child health, control of infectious diseases, and health education, but do not receive salaries for these services. CHWs make up the largest single group involved in health delivery in the country; however, limited information is available regarding the socio-economic circumstances and satisfaction levels of this workforce. Such information can support governments aiming to control infectious diseases and alleviate poverty through enhanced healthcare delivery. The objectives of this study were to (1) evaluate CHW opportunity costs, (2) identify drivers for CHW motivation, job satisfaction and care provision, and (3) report CHW ideas for improving retention and service delivery. In this mixed-methods study, our team conducted in-depth interviews with 145 CHWs from three districts (Kirehe, Kayonza, Burera) to collect information on household economics and experiences in delivering healthcare. Across the three districts, CHWs contributed approximately four hours of volunteer work per day (range: 0–12 hrs/day), which translated to 127 684 RWF per year (range: 2 359–2 247 807 RWF/yr) in lost personal income. CHW out-of-pocket expenditures (e.g. patient transportation) were estimated at 36 228 RWF per year (range: 3 600–364 800 RWF/yr). Participants identified many benefits to being CHWs, including free healthcare training, improved social status, and the satisfaction of helping others. They also identified challenges, such as aging equipment, discrepancies in financial reimbursements, poverty, and lack of formal workspaces or working hours. Lastly, CHWs provided perspectives on reasonable and feasible improvements to village-level health programming that could improve conditions and equity for those providing and using the CHW system.

to H.A. Website: https://www.cummingsfoundation.org/ The funders had no role in study design, data collection and analysis, decision to publish, or preparation of the manuscript.

**Competing interests:** The authors have declared that no competing interests exist.

## Introduction

Community Health Workers (CHWs) are citizens with no formal medical education who have been elected to provide basic health services in the villages where they reside [1–3]. Recruitment and training of citizens to provide primary care (e.g. vaccination) originated in China in the 1920s and has since expanded to a many other countries, including almost all African member states, with wide variation in job titles and program descriptions [1–3]. In 2019, a United Nations declaration reaffirmed health as a foundational requirement for sustainable development and set a goal to achieve universal health coverage by 2030 [4]. Significantly, the declaration committed to supporting CHWs and their patients by scaling up training, increasing retention, and improving service delivery for underserved populations through worker incentives [4].

In Rwanda, CHWs were first deployed in 1995 at a time when the health infrastructure was in disarray, health professionals were in short supply, and the average life expectancy was 31 years [5, 6]. Nearly 25 years later, life expectancy has more than doubled (68 years), but physicians remain in short supply with only 0.1 physicians and 0.9 nurses/midwives per 1000 people, well below the threshold established by the World Health Organization [6–9]. CHWs have become an integral part of the Rwandan health system, linking even the most remote households to primary care, and are credited with significant gains in reducing maternal and under-five mortality, reducing the burden of infectious diseases such as HIV/AIDS, tuberculosis and malaria, increasing vaccine coverage, and improving access to family planning [1, 5].

Worldwide, attrition of health workers, including CHWs, is an obstacle to providing sustainable and quality health services [3, 10]. Human resource losses disrupt patient-provider relationships and increase health system expenditures, as new workers must be adequately trained and equipped prior to deployment [3, 11]. In Rwanda, both CHWs and patients report high levels of satisfaction in village-level care, but despite this, up to 10% of CHWs leave the program per year [11, 12]. Major drivers for attrition in Rwanda, and elsewhere, include issues with financial compensation and workload [12–14]. In Rwanda, CHWs are unpaid, although they can receive monetary incentives as part of a performance-based financing (PBF) scheme introduced in 2008 that rewards both quantity and quality of services related to specific health indicators (e.g. TB, HIV, stunting) [9]. Monetary incentives are transferred to CHW cooperatives that invest 70% in income generating activities (e.g. livestock production) and distribute the remaining 30% among their members [5]. CHWs are meant receive personal items (e.g. gumboots, raincoat, bag, torch) and supplies for treating patients (e.g. cup, cupboard, cellular phone, rapid diagnostic tests, mid-upper arm circumference tape) [11].

Countries around the world have recognized long-term retention of skilled workers as a key indicator for health systems performance and optimal program financing [3, 10]. The Rwanda Ministry of Health (MOH) Strategic Plan for 2013–2018 aimed to improve delivery and utilization of health services and to strengthen CHW cooperatives through monetary and non-monetary incentives [5]. It identified knowledge gaps on how CHWs spend their time as a key barrier to optimizing the benefits of community service and income generation. A better understanding of CHWs resource outputs, motivations and challenges in providing care could help Rwanda and other low resource countries move strategically towards universal health coverage. Therefore, the aims of this study were to (1) calculate the opportunity costs incurred by CHWs, (2) characterize factors contributing positively and negatively to CHW motivation, job satisfaction and service delivery, and (3) describe CHW perspectives on strategies for improved satisfaction and service delivery.

## Methods

### Study setting

Rwanda is a low-income country in East Africa with a population of approximately 12.5 million people [15]. It has the second highest population density in Sub-Saharan Africa (499 individuals/km$^2$) and the population is skewed towards younger age categories (median age—19.6 years) [16, 17]. In 2018, the gross domestic income per capita was 772 750 RWF (826 USD), with most residents (72%) reliant on agricultural activities for income [18, 19]. Infectious diseases, such as lower respiratory infections, tuberculosis, diarrheal diseases, malaria and HIV/AIDs, are among the top ten causes for mortality [20]. Approximately 86% of households purchase Community-Based Health Insurance (CBHI) to cover healthcare costs [21].

The Rwanda MOH has invested significantly in village-based health programming as a measure to fill the gap in skilled health providers, to improve patient equity and to expand access for rural residents (Fig 1) [5, 9, 11]. CHWs are elected and must meet basic criteria such as willingness to volunteer, age 20–50 years, ability to read and write, living in the village, and being regarded as honest, trustworthy and reliable by fellow residents [5]. Three CHWs work in each village of 50–150 households: a male-female pair (agents de santé binôme) that focuses on integrated community case management of childhood illnesses, and one Animatrice de Santé Maternelle (ASM) who focuses on maternal and newborn health services [5]. Together, they are responsible for a comprehensive range of services, including health education, malnutrition screening, family planning, and infectious disease management (e.g. diarrhea, pneumonia, malaria, and tuberculosis) [5].

### Study design

This mixed-methods, cross-sectional study was conducted in three of the 30 districts in Rwanda. Malaria case management is rewarded by PBF and a significant contributor to CHW

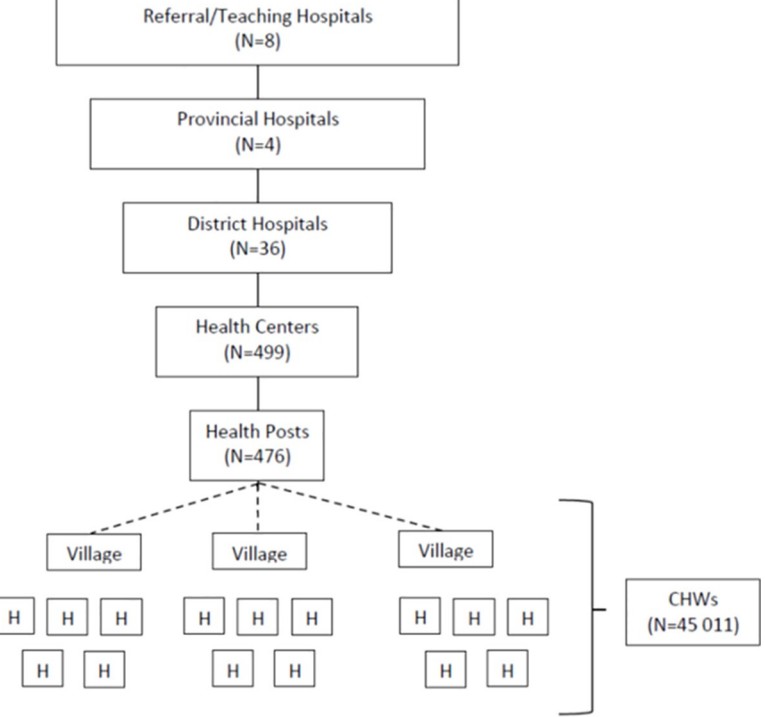

**Fig 1. Rwanda health system organizational structure (adapted from [9]).**

workload. Therefore, we selected three districts in Rwanda based on malaria incidence (low-Burera, moderate—Kirehe, high—Kayonza), using this as a proxy for differences in time and monetary outputs among CHWs (Fig 2) [22]. The team developed a three-part questionnaire that used closed-ended questions to obtain demographic information and time/monetary outputs, and open-ended questions to obtain perspectives on benefits, barriers, and retention strategies. Data collectors probed open-ended responses to clarify meaning where needed. The questionnaire was translated from English to Kinyarwanda and then back translated to ensure content validity. It was pre-tested with four CHWs outside the study region and revised as needed.

Approximately 45 516 CHWs work in Rwanda and we estimated that 4 812 CHWs currently work in the three study districts [3 workers/village * (421 + 612 + 571 villages)] [9]. Therefore, potential study participants in the target districts made up 10% of the national population of CHWs. We used the following formula to determine sample size:

$$N = \frac{z^2 * \hat{p}\,(1-\hat{p})}{\varepsilon^2}, \text{ where } z = 1.96,\ \hat{p} = 0.1,\ \text{and } \varepsilon = 0.05$$

Consequently, our goal was to recruit 138 CHWs to participate in the study. To ensure that we reached this goal, we invited a total of 144 participants to participate. Within each study district, we targeted the health center with the highest number of CHWs, obtained lists of all CHWs separated by gender, and randomly invited equal numbers of women and men to participate.

The study team was comprised of researchers from Rwanda, Canada, and the USA, covering expertise in a wide range of disciplines including health economics, human medicine, gender studies, community-based programming, and global health. Rwandese data collectors received training on the questionnaire before conducting one-on-one interviews with the selected CHWs in private areas at each health center. Responses to open-ended questions were audio-recorded with permission from participants. Data related to demographics and time/monetary outputs were entered into a spreadsheet and checked for errors. Demographic

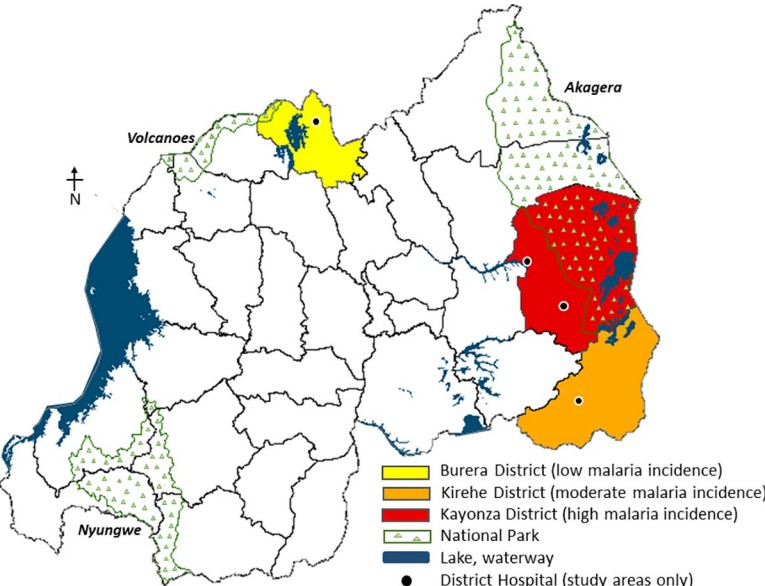

**Fig 2. Map of Rwanda illustrating the three districts selected to participate in this study (malaria incidence as per [22]).**

differences between study participants in the three study sites were evaluated in SPSS statistical software (version 25; IMB, Chicago, IL) using the Fisher Exact Test with a 2-sided p-value threshold for statistical significance set at 5%.

## Opportunity cost analysis

Time/monetary outputs obtained from the questionnaire provided the data to estimate opportunity costs for each CHW in our sample, and with this, the median (interquartile range) opportunity cost by district and across all three districts (Table 1). To determine the total household earnings per day for each CHW, we estimated cash earnings (i.e. salary, cash benefits) and *in-kind* income (i.e. goods produced from labour) not related to community health work (Equation 1). To calculate *in-kind* income, we converted goods produced, in this case agricultural products, to a monetary value using the average market price of these goods from 2018, as reported in e-Soko [23]. All market prices were adjusted to 2019 RWF. Moreover, we calculated total household number of working hours per day (Equation 2), average income per hour (Equation 3), and the number of hours spent on CHW activities in a day, averaged over a twelve-month period (Equation 4). We then calculated opportunity costs for each CHW by multiplying CHW income per hour by the number of hours worked per day on CHW activities (Equation 5).

## Thematic analysis–CHW motivations, job satisfaction and care provision

Audio-recordings capturing perspectives on benefits, barriers, and retention strategies were transcribed in Kinyarwanda, translated into English, and then reviewed for accuracy against

**Table 1. Equations used to calculate Community Health Worker (CHW) opportunity costs[1] in Rwanda.**

| | |
|---|---|
| **Equation 1[2]** | ***Household income per CHW per day*** = $\left(\dfrac{\textit{cash income per CHW per month} +}{\textit{inkind income per CHW per month}}\right) \Big/ 30 \textit{ days}$ |
| **Equation 2** | ***Household working hours per day*** = $\textit{CHW income generating hours worked per day} + \left(\dfrac{\textit{number of working nonCHW household members} \times}{\textit{median hours worked per day}}\right)$ |
| **Equation 3** | ***Income per hour*** = $\dfrac{\textit{household income per CHW per day}}{} \Big/ \textit{household working hours per day}$ |
| **Equation 4[3]** | ***Number of hours of CHW work per day*** = $\left(\dfrac{\textit{number of busy months} \times}{\textit{hours per day worked in busy months}}\right) + \left(\dfrac{\textit{number of normal months} \times}{\textit{hours per day worked in normal months}}\right) + \left(\dfrac{\textit{number of least busy months} \times}{\textit{hours per day worked in least busy months}}\right) \Big/ 12 \textit{ months}$ |
| **Equation 5** | **Opportunity costs per CHW per day** = Income per hour × number of hours of CHW work per day |

[1]We calculated the opportunity costs for each individual CHW and used those values to estimate the median opportunity costs across all CHWs.

[2]*In-kind* income represents the goods produced from labour that can be sold for income (in this case, mostly cash crop).

[3]We averaged the hours worked per day in the least busy, moderate and most busy month, while accounting for the number of months on average the CHWs worked in each category per year.

the original transcript by a second translator. Thematic analysis of the qualitative data collected on CHW perspectives was conducted using the constant comparative method. Three researchers (KF, JM and JMS) worked as a team to inductively develop the codebook. A draft codebook was developed after reading the first 20 transcripts and then modified throughout the coding process to include emerging themes. Inter-operator variability was compared to detect differences in sentence coding after all transcripts were coded using Cohen's Kappa with an agreement threshold of 0.8 per code. Coding and comparison were conducted using NVivo (version 12; QSR International, Melbourne, Australia).

### Ethics

This research was reviewed and approved by the Institutional Review Boards at Tufts University (#1955026) and the University of Global Health Equity (#0084). Further approval was obtained from CHW managers in the three study districts.

## Results

### Study participants

In total, 145 CHWs from the three districts participated in our study, including one participant who volunteered in addition to those originally invited (participation rate = 100%). Participants were evenly distributed among males and females, most commonly aged between 31 and 50 years, and usually married (Table 2). Most were binômes (80%) as opposed to ASMs (20%). Occupation was the only variable that differed significantly between locations (p = 0.029); although farming (cash crop or livestock production) remained the most common primary source of income for CHWs in each location. On average, CHW households were comprised of six family members (± 1.7), including three school age children (±1.6) and two income generating adults (±0.8). Study participants had worked as CHWs between zero and 19 years, with half reporting 10–14 years of experience (average 9 ± 3.7 years)

### Opportunity cost analysis

When asked about the seasonal differences in their activities, CHWs reported being most busy for four months (±2) of the year, during which they expended a median of five hours per day on CHW activities (Table 3). They were least busy for three months (±1.5) of the year when they contributed a median of two hours per day on CHW activities. CHWs contributed a median four hours per day during the remaining five months of the year. In total, CHWs worked an average of four hours per day across the year. In addition to these time expenditures, participants worked a median of six hours per day on activities unrelated to their CHW duties (e.g. farming) throughout the year.

At the household level, CHWs reported a median annual income of 358 128 RWF (range: 24 000–7 611 707 RWF), including direct monetary income as well as the value of agricultural goods produced on household farms (Table 4). Across the three districts, each CHW donated a median of 127 684 RWF (range: 2 359–3 247 807 RWF) annually in opportunity costs related to CHW activities. Participants estimated that they also contributed a median of 36 228 RWF (range: 3 600–364 800 RWF) per year out-of-pocket on activities related to patient care. These expenditures primarily covered transport for CHWs and their patients, as well as food during training sessions, cellular airtime, patient medication, soap, and supplies for expecting mothers. In return, CHWs received gratitude and respect from patients and community leaders, but no monetary gifts. Sometimes, CHWs received food or invitation to a child christening.

**Table 2. Demographic summary of study participants engaged in community health work in three districts of Rwanda (N = 145).**

| Variable | Number (%) | | | | Fisher's Exact Test P-value |
|---|---|---|---|---|---|
| | Burera (n = 49) | Kayonza (n = 48) | Kirehe (n = 48) | Total (N = 145) | |
| Sex | | | | | |
| Female | 21 (43) | 24 (50) | 24 (50) | 69 (48) | 0.729 |
| Male | 28 (47) | 24 (50) | 24 (50) | 76 (52) | |
| Age (years) | | | | | |
| 20–30 | 3 (6) | 1 (2) | 4 (8) | 8 (6) | 0.768 |
| 31–40 | 13 (27) | 19 (40) | 15 (31) | 47 (32) | |
| 41–50 | 20 (41) | 17 (35) | 18 (38) | 55 (38) | |
| >50 | 13 (27) | 11 (23) | 11 (23) | 35 (24) | |
| Marital Status | | | | | |
| Married | 48 (98) | 46 (96) | 43 (90) | 137 (94) | 0.185 |
| Divorced | 0 (0) | 1 (2) | 0 (0) | 1 (1) | |
| Widowed | 1 (2) | 1 (2) | 5 (10) | 7 (5) | |
| Occupation | | | | | |
| Farming | 46 (94) | 48 (100) | 45 (94) | 139 (96) | 0.029 |
| Self-employed | 3 (6) | 0 (0) | 0 (0) | 3 (2) | |
| Unemployed | 0 (0) | 0 (0) | 1 (2) | 1 (1) | |
| Other | 0 (0) | 0 (0) | 2 (4) | 2 (1) | |
| CHW type | | | | | |
| Binôme | 40 (82) | 41 (85) | 35 (73) | 116 (80) | 0.324 |
| ASM[1] | 9 (18) | 7 (15) | 13 (27) | 29 (20) | |
| CHW length (years) | | | | | |
| 0–4 | 4 (8) | 9 (19) | 1 (2) | 14 (10) | 0.198 |
| 5–9 | 14 (29) | 14 (29) | 19 (40) | 47 (33) | |
| 10–14 | 26 (54) | 23 (48) | 26 (54) | 75 (52) | |
| ≥15 | 3 (6) | 2 (4) | 2 (4) | 7 (5) | |
| Missing[2] | 2 (NA) | 0 (NA) | 0 (NA) | 2 (NA) | |

[1]ASM—Animatrice de Santé Maternelle

[2]Percentage calculated by removing missing values from denominator

### Thematic analysis–CHW motivations, job satisfaction and care provision

Our analysis identified four recurring themes essential to understanding the experiences and perspectives of CHWs providing village-level health services:

1. CHWs are primarily motivated by political solidarity, training opportunities, and the desire to help others by improving health services

2. Family support strongly influences CHW satisfaction and retention

3. Uncompensated work impedes improvements to household standard of living

4. Programmatic gaps prevent CHWs from optimizing service delivery at the village level

**1) CHW Motivation.** Our study participants were strongly motivated to become CHWs by a sense of personal and political responsibility to help vulnerable people, serve their communities, and develop their country. Being a CHW was perceived as a good way to network with other community members or healthcare workers, to join a respectable group, and to rise

**Table 3. Summary of time outputs reported by participants on Community Health Worker (CHW) activities per person per day in three districts of Rwanda.**

| | Burera (N = 49) | Kayonza (N = 48) | Kirehe (N = 48) | Overall (N = 145) |
|---|---|---|---|---|
| **CHW work (hours/day)[1]** | | | | |
| Least activity months | | | | |
| Median | 3 | 3 | 2 | 2 |
| Range | 1–6 | 0–6 | 0–6 | 0–6 |
| Moderate activity months | | | | |
| Median | 4 | 4 | 3.5 | 4 |
| Range | 1–12 | 2–11 | 1–8 | 1–12 |
| Busiest activity months | | | | |
| Median | 6 | 6 | 4 | 5 |
| Range | 2–12 | 0–12 | 0–12 | 0–12 |
| **Non-CHW work (hours/day)** | | | | |
| Median | 7 | 5 | 6 | 6 |
| Range | 3–12 | 1–7 | 2–14 | 1–14 |
| **Annual median work (hours/day)[2]** | | | | |
| CHW work | 4.7 | 4.8 | 3.8 | 4.2 |
| Total work | 10.7 | 9.0 | 10.3 | 9.9 |

[1]CHW were least busy three months of the year, moderately busy five months of the year, and most busy four months of the year.

[2]Median of hours/day across all 12 months

**Table 4. Summary of Community Health Worker (CHW) income, monetary outputs and opportunity costs in three districts of Rwanda (2019RWF, 2019USD).**

| | Burera (N = 49) | Kayonza (N = 48) | Kirehe (N = 48) | Overall (N = 145) | |
|---|---|---|---|---|---|
| **Currency** | —————————RWF/year————————— | | | | USD/yr |
| **Median CHW household income** | | | | | |
| *Direct monetary payments* | 120 000 | 60 000 | 36 000 | 80 000 | 86.49 |
| *In-kind income[1] (agricultural goods)* | 393 748 | 328 017 | 307 285 | 334 300 | 361.41 |
| Total[2] | 630 927 | 478 392 | 475 763 | 531 010 | 574.06 |
| Min (Total) | 84 259 | 66 149 | 24 000 | 24 000 | 25.95 |
| Max (Total) | 7 611 707 | 3 842 074 | 6 064 152 | 7 611 707 | 8228.87 |
| **Median CHW income** | | | | | |
| Median | 250 690 | 149 570 | 207 230 | 205 505 | 222.17 |
| Min | 32 548 | 23 695 | 4 923 | 4 923 | 5.32 |
| Max | 1 880 539 | 1 948 684 | 1 534 304 | 1 948 684 | 2106.69 |
| **CHW opportunity costs** | | | | | |
| Median | 150 377 | 137 140 | 94 702 | 127 684 | 138.04 |
| Min | 22 571 | 10 696 | 2 359 | 2 359 | 2.55 |
| Max | 2 514 290 | 3 247 807 | 2 837 716 | 3 247 807 | 3511.14 |

[1]*In-kind* income represented agricultural goods produced by labour that could be sold for income

[2]Total represents the median of all the individual CHW household income totals and therefore does not equal the median direct monetary payments plus the median *in-kind* income.

in social status. Being trusted to deliver health services was another powerful motivator. While some had sought election, others were nominated and felt obligated to accept and earn the trust demonstrated by fellow community members. According to CHWs:

> "The major reason why I became a CHW is because of the need to help the community. I was passionate to help my country."

> "I realized that someone like a CHW is lovable and very respected in society. By now, people respect me and they want to talk with me. No one will talk to me rudely or do anything bad to me."

> "I became a CHW because of the love for my villagers, to be trusted by villagers and chose you to help them is something valuable."

The promise of free healthcare training was another strong motivator for accepting the role of CHW. Some considered CHW training to be a good substitute for unfulfilled school goals, while others thought that being a CHW might ultimately lead them to a higher level position such as nursing.

> "I used to wish to become a nurse, but there were some beliefs that triggered my parents to think that it is not worth spending money on studies for a girl like me. I felt so angry for some years because they used to undermine girls. So, I realized that my dreams would come partially true when I got the chance of being a CHW. That is the reason why I am a CHW and I love it."

Participants spoke about the realities of living in rural villages–infectious diseases such as malaria that caused death and illness, treatment delays such as long distances to health centers or long line-ups to see physicians, and poor knowledge of preventative measures such as antenatal care or clinic deliveries. They felt a personal responsibility to use their CHW training to disseminate knowledge, promote healthy behaviors, prevent/treat illnesses, and improve access to care. Some CHWs expressed pride in contributing to a program that led to improved maternal and child health and reduced infectious disease.

> "It is because I wanted to be knowledgeable about all health-related things in order to take care of people's health, including mothers and children at home. If I didn't become a community health worker, I wouldn't be aware of anything related to health."

> "The reasons why I am happy working as a Community Health Worker: First, it is a means of helping people who were having to travel a long distance to health centers to seek services. Before, an infected child might have to spend a long period of time without treatment. Then, we decided to be volunteers and can help people in our villages."

Although many become CHWs knowing that they would be volunteers, others were motivated by the belief that their quality of life would improve through incentives or financial remuneration.

> "We heard that CHWs do get something and that motivated me to enroll as a CHW. Unfortunately, I have never seen or received any benefits."

**2) Family support.**    Family backing, or lack thereof, was strongly associated with CHW satisfaction and retention. Many who experienced positive support had sought their spouse's

backing prior to joining the program. For others, support grew once a family observed the CHW treating sick patients or when they personally benefitted from CHW services. This translated into family members helping to write reports, reminding the CHW about meetings, bringing them a ringing cell phone, and not complaining when the CHW was called away at an inconvenient time.

> "They are okay with it, because they benefit from my knowledge. For instance, if a child falls sick, I can treat him/her right away. If my family finds me treating sick children for our neighbors, they appreciate that too."

Other families were less supportive and CHWs identified familial support as a key factor in their continued involvement. Complaints occurred when spouses were left alone to complete farming or other household activities, when patients arrived at the CHWs home requesting assistance, and when children went unfed because their parent was called away. Some CHWs described situations where they had to leave spouses to cultivate crops alone because they were called away to assist a patient and this led to strained relationships. Most frequently, family members objected to the long hours CHWs spent away from home with no financial compensation. Some CHWs expressed shame that they were unable to contribute financially to their family's wellbeing.

> "Difficulties are limitless. First, in a few minutes it will be lunch time. I came here in the morning and I left home without cooking. I am pretty sure that my child who is supposed to go back home from primary school at noon won't get anything to eat and will return back to school without eating."

**3) Standard of living.**   Overall, participants felt strongly that CHWs performed valuable and time-consuming work that deserved a monthly salary. Household poverty was frequently mentioned as a constraint on family wellbeing, a reason for resigning from CHW activities, and a barrier optimizing service delivery. Some CHWs were already poor when they joined the program, and some became even poorer due to out-of-pocket expenses that were not reimbursed. Poverty among patients exacerbated this financial insecurity when CHWs covered the expenses of patients who could not afford transport fees or treatment costs.

> "I thought that I would get some money-oriented benefits from this job, but unfortunately I haven't gotten anything. I thought that I would able to contribute to the development of my family. Discordant to my expectations, I am now losing money, even what my home gains, in transportation."

> "They [CHWs] resign because they are poor and don't get money or anything needed to respond to their families' needs. This is mostly the underlying reason for their resignation."

CHWs described feeling tired due to the cumulative responsibilities of CHW and household work. Time and energy spent on CHW duties interfered with domestic obligations (e.g. farming, caring for children) and also limited participation in income generating activities. Furthermore, CHWs experienced exhaustion after walking long distances to perform their duties.

> "Another challenge is in cultivating season where I don't get the time to cultivate Irish potatoes for money because I am busy with CHW activities. This can delay agricultural

activities, thus poor harvest. I wish we could get money for these CHW activities to finance our agricultural activities."

"Sometimes we have trouble in the family. For example, if I plan to accept a job I am offered and make 1000 RWF so that we can buy food, but I get a call saying that someone is seriously ill, I immediately forget about making money and go take care of him or her. You can understand the problem. My family says, "You see, he is going, while he would make money for food tonight.""

"Another challenge is when we go to see a patient who is far away and we starve and get tired, because we have no money for food or transportation."

An increase in CHW responsibilities has exacerbated the lack of personal time for some CHWs. According to some CHWs, they were until recently only responsible for the care of children under 5 years of age. Now their scope of care also includes adults, which requires a broader range of expertise, from family planning to common infectious disease diagnosis and treatment.

"We face many challenges. The main one is that we have been given more tasks to complete. We used to treat children under five years of age, but now we are treating adults, too. In this way, our responsibilities have increased and we don't receive any support."

Discrepancies were evident in the expectations and realities of financial reimbursements. While many CHWs understood that they were being recruited for a volunteer position, some believed they would receive salaries. Sometimes, study participants spent the last of their money on CHW-related costs (e.g. transportation) and were then unable to buy household items, pay school fees, or cover CBHI fees. Some participants reported that they had previously received monthly stipends, but that these stipends had stopped; others described no interruptions in reimbursements. Differences were also present regarding the frequency and amount of PBF received, with some CHWs stating that they had never received payments. The false perception among friends and neighbors that CHWs received salaries made this issue more painful.

"Recently, we were supposed to get 8000 RWF in incentives, from which they would take away 1000 RWF for the cooperative. After three years, we got 5000 RWF. Thereafter, it was reduced to up to 1000 RWF, which is what we get now. It seems like we do not have incentives. We were given this money twelve times, because it was given monthly and we had it for one year. As I told you, I have been working as a CHW for three years."

"Sometimes there were rumors that government will pay insurance for CHWs or any other support, but we didn't receive any of this support. We are not blaming government, but I have not received anything from the government for this job."

CHWs and their families benefited from CHW work through heightened social status and acquired health knowledge, overall leading to better standard of living. Many stated that these benefits were sufficient motivation to keep them providing village-level care, regardless of financial compensation.

"They appreciate my involvement because there are considerable changes at home that I have made, such as hygiene and family planning."

**4) CHW program challenges.** CHWs expressed a high level of satisfaction and pride in their work but were frustrated by certain programmatic challenges that impeded their performance and/or patient experiences. These included delays in receiving equipment and supplies, poor equipment quality, such as solar panels with limited light and poor durability, or equipment that had become old and not been replaced. CHWs greatly appreciated boots and raincoats for the rainy seasons, solar panels that allowed them to write reports and treat patients in the evenings, cell phones, and personal care items (e.g. soap and cloth) that would help them meet the CHW hygiene standards. A few suggested that providing smart telephones would allow them to submit reports by internet and avoid transportation fees associated with travel to the health clinics.

> We used to receive 5000 RWF as a compensation fee and we got this for many years, but it was finally removed. I do not remember well the number of years I received it but it was for more than seven years. The Ministry of Health used to help us in relation to our job by giving us equipment like boots, umbrellas, lamps and other things like that. They gave us boots twice, umbrellas twice, and lamps once. They also gave us a solar panel but now we do not get anything. The boots and umbrellas we have are old as they have lasted us six or seven years.

In addition to supply shortages, CHWs described the difficulties of having no formal space for treating patients in their villages. CHWs stored medical supplies, equipment and documents in their homes, risking damage by water or rodents; the lack of secure medicine cabinets was a concern for parents of small children. Binômes received storage cupboards, but ASMs did not. CHWs commonly used their homes to treat patients and expressed the need to have dedicated furniture (tables, chairs) for patients. Patients showed up for care at the CHW's home unexpectedly and this led to awkward encounters if the family had just sat down to eat a meal or if they were hosting visitors. Such situations eroded the boundary between personal and professional life and created challenges for maintaining patient privacy. One participant stated that a formal space could help CHWs establish regular working hours and would improve the overall patient experience.

> "The patients might come to my house while the food is already served. I can disturb others while I am bringing seats to the patients. I may take my husband's chair when he wanted to use it while eating. So, it is a matter of inadequate workspace."

> "Another challenge is where we keep medication. Some of us have children and keeping medicines where they can come in contact with them isn't really safe."

> "Sometimes, I have visitors and a patient comes, and there is no way I can treat a patient while respecting his or her privacy. For example, privacy is a necessity when providing family planning services, but we do not have ways to provide it to our patients because we do not have enough space."

The desire to receive education was a top reason for becoming a CHW. Many felt increasing the frequency of trainings would improve their job performance, and some mentioned that they had yet to be trained for their role.

Transportation was one of the most common programmatic issues mentioned. Participants regularly walked long distances to reach patients at their homes, to accompany patients to the health center, and to reach the health center for monthly meetings. Walking alone at night or during the rainy season was difficult and led some to feel unsafe. Many spent their own money

on transportation, sometimes paying for patients who could not afford the fees. Often reimbursement was insufficient, delayed or absent. Many CHWs felt that a bicycle could help them to reach patients faster and to attend meetings on time.

> There are difficulties when you get a call at night that there is a woman in labour and you live in a district where there are stones from volcanic eruptions. Then, you have to go by foot without shoes, a torch, or clothes which would protect us against the cold. We wish to be given bicycles to make the journeys easier. A motorbike in every cell would help CHWs when there are quick reporting or when we go to meetings.

Those receiving remuneration felt that the PBF process could be improved. Most importantly, CHWs wanted to receive PBF directly instead of losing money to cooperatives which were sometimes mismanaged or unproductive. Some stated that the current process of being reimbursed through Umurenge Sacco's (Savings and Credit Cooperatives) meant lost money through bank fees. One responsibility of CHWs was to encourage community members to pay health insurance (CBHI) fees each year. Some were grateful to receive CBHI for their families through the CHW cooperative. Others were not covered by their cooperative and felt embarrassed encouraging community members to pay CBHI fees when they themselves could not afford to pay.

> Another challenge is when cooperative money is mismanaged, and we miss profits. You find at the end of year there is no significant profits which should be significant in case there is appropriate cooperative money management.

> "We also have a difficulty about the money of PBF, they give us 30% and another 70% is transferred into our cooperative and that money that goes in our cooperative does not help us. And the 30% we get is passed through the Umurenge Sacco bank and if we were to take 1500 RWF or 2000 RWF they deduct it and we receive almost nothing which cannot help us and when we have used transportation tickets."

> "Sometimes we go to the health center when they have arranged the transportation reimbursement of 2000 or 3000 RWF to be got from Sacco bank. And I use 2000 RWF to and from the health center and I will again use other 2000 RWF going to Sacco for withdrawing and coming back at home and the total is now 4000 RWF and when I reach to Sacco they deduct 500 RWF from the 3000 RWF that I was supposed to receive and I remain with 2500 RWF when I have spent 4000 RWF and that is a big challenge to us because sometimes we do not have courage for bringing that money because we spend a lot and they do not give that money in hand."

Study participants were aware of CHWs who had left the program. Reasons for resigning included moving to a new community, obtaining a paid job, becoming ill or needing to care for an ill family member, feeling too old to perform all duties, lack of financial compensation or family support, and feeling shame when a family member did adhere to CHW protocols. Other CHWs were terminated involuntarily due to misconduct, poor reporting, failing to exemplify proper behavior, alcoholism, and unsatisfactory work performance.

## CHW ideas for improving satisfaction and service delivery

In general, CHWs expressed great love for performing their duties and were appreciative of incentives provided by their government. CHWs were practical in their suggestions for incentives, knowing that Rwanda is a low income country and that there are many CHWs. While

**Table 5. Community Health Worker suggestions for improving patient and provider satisfaction.**

| Category | Improvement |
|---|---|
| Compensation | • Provide monthly salaries or waive CBHI/school fees<br>• Pay PBF incentives directly (avoid SACCO fees)<br>• Domestic livestock (e.g. cow) to combat malnutrition (milk, manure) |
| Transportation | • Provide bicycles or motorbikes<br>• Timely reimbursement of travel expenses |
| Equipment | • Resupply at regular intervals<br>• Raincoat, rubber boots, smartphones, torches |
| Infrastructure | • Formal treatment space with tables, chairs and toilet in village<br>• Secure medicine cabinet, waterproof bag to carry and store documents |
| Training/Career | • Increased training sessions<br>• Opportunity to move up the health provider ladder (e.g. become a nurse) |

many hoped to receive monthly salaries, other were happy to receive a few dollars each month to cover transportation costs, CBHI, or school fees. CHWs provided suggestions that they felt would help them improve service delivery, either through increased efficiency or increased retention of skilled workers due to job satisfaction (Table 5).

## Discussion

At a time when CHW programs worldwide are being scaled up and promoted as a key building block in the fight for equity in access to healthcare, we provide insights into the Rwandan program, which has been recognized for extraordinary achievements in population-level health [1, 5]. These improvements to key health indicators occurred in spite of chronic physician shortages and were accompanied by high public satisfaction ratings for CHW services, signifying an opportunity to employ this program for other emerging health priorities [7, 11, 12]. According to our calculations, CHWs each contribute 28 hours of unsalaried CHW work per week, translating to 137 USD in lost opportunity costs per year or 62% of an average CHW income [18, 19]. Our estimate was slightly higher than the 20 hours/week (range 2–68 hrs/week) reported in 2016 [12], which is explained by recent increases in workload. CHWs also contributed approximately 18% of their income to costs associated with patient care (e.g. patient transportation) and reported programmatic inconsistencies with remuneration. Discussions on shifting tasks from formal healthcare workers to CHWs should consider current responsibilities to ensure that adequate provisions are in place to avoid overburdening a group already struggling with workload.

CHWs deliver a broad range of health services at a fraction of the costs incurred for care provided by physicians and nurses [2]. Across all sites, CHWs in Rwanda consistently expressed the desire for reasonable financial compensation. This desire for salary and/or transport reimbursement is shared by village health workers worldwide [13, 24–27] and is not surprising given that CHWs are often from poorer socioeconomic groups, lacking education and professional opportunities to enhance their standard of living. Introduction of a government salary has been demonstrated to reduce attrition up to 50% when compared to CHWs who rely on community financing [14]. In Rwanda, PBF is linked to specific health indicators and CHWs are supposed to receive 30% of this incentive; however, some CHWs reported that they had received no payments. The current process of re-directing 70% of PBF through cooperatives and Sacco's was a clear source of frustration, and if possible, could be restructured to improve levels of satisfaction.

Despite these challenges, many study participants stated that they would continue to provide services regardless of financial benefit. CHWs were strongly committed to providing

village-level care and were motivated to serve their country, to help vulnerable individuals, and to strengthen their health system. CHWs valued free health training and were encouraged by the gratitude and respect that they received from their communities. These findings were supported by another recent study in Rwanda which reported high satisfaction (98%) among CHWs and three key drivers for motivation–social status, training and desire to help [12]. Similar findings have been reported among CHWs in Tanzania and Uganda [28, 29]. Understanding the factors that motivate CHWs is a key step to improving service delivery and to retaining a skilled and trusted health workforce.

Our study participants were primarily farmers aged 31 to 50 years who worked six hours per day performing manual labor on their plots and four hours per day on CHW activities. These long hours resulted in fatigue as well as family tension, especially when CHWs were interrupted from household duties such as meal preparation or when spouses were left alone to complete farming activities. Fatigue was commonly linked to travel, as CHWs walked long distances to health centers or to find patients, sometimes at night or in the rain. One recent study estimated that 25% of Rwandese CHW time outputs were spent in transit [12]. Inconsistent remuneration for commercial transport (bicycle taxis, motorcycle taxis or minibus) discouraged CHWs from using these options. Furthermore, spending personal finances on transportation led to conflict in some households if this meant no money remained for food, school fees or health insurance. Our participants consistently requested support in advocating for transportation solutions, ranging from bicycles and motorcycles to direct cash transfers. Transportation challenges were also raised among CHWs operating in other African countries, including Mozambique, Uganda, Kenya and Tanzania [13, 24–27, 30]. Supporting CHW transport needs would clearly reduce time outputs and fatigue but no studies have identified the most efficient and preferred intervention.

Most (95%) study participants had been CHWs less than 15 years, suggesting regular turnover of the workforce. Reasons for attrition fell into two broad categories also identified by other studies in Rwanda—personal (e.g. lack of family support or relocation) and programmatic (e.g. insufficient training/equipment/supplies, inconsistent supervision, variable and unpredictable workload, and inadequate compensation) [11, 12, 31]. CHWs in Uganda reported similar reasons for attrition, including age, relocation, termination, need for income [29]. Our conversations with CHWs highlighted the significance of family support for retention, especially as many stated that they would quit the program if their spouse became dissatisfied with their volunteerism. Similar to other studies, family members were frustrated by lost income, patients interrupting meal-times, travel away from home, and being left alone to farm or care for children [12, 31]. CHWs reported that discussing the program with a spouse prior to becoming a CHW was a good strategy for maintaining harmony. Furthermore, family members were more willing to make sacrifices on their behalf when they benefitted personally or when they observed others being helped by CHWs.

Despite our best efforts, this study had several limitations. First, all financial data and perspectives on CHW programming were self-reported. Similar values for reported income and out-of-pocket spending between study locations and the saturation achieved on qualitative data suggest that study participants characterized their situations truthfully. Furthermore, our calculations used median values to minimize the effect of outliers. Second, we used a national tool (e-Soko) to estimate value for agricultural goods produced by CHW households and this did not account for regional price differences. Third, our characterization of PBF was limited to qualitative data and did not allow for precise calculations which would be necessary to compare between groups. Lastly, our calculation of opportunity costs did not account for PBF received. Our interviews with CHWs suggested significant differences in PBF benefits received and this issue should be explored in a follow-up study.

This study demonstrated tremendous commitment among CHWs for serving their communities and strengthening the Rwandan health system. The mixed-method design increased our understanding of CHW time and financial contributions and provided evidence to support appropriate investments in a workforce that is essential to achieving equitable access to health services. CHWs in this study were not solely motivated by financial gain. Many participants provided ideas for non-monetary incentives (e.g. bicycles, waiving school fees) that could increase motivation, reduce attrition, and enhance service delivery. Such investments in CHW retention are justified given the low-cost of a program that has a proven track record of mitigating the negative consequences of health worker shortages. Moreover, these investments could help reduce the financial losses incurred for training/re-training workers and ensure continuity of care from skilled and trusted health workers, while improving program effectiveness [14]. Rwanda is a low income country, and despite consistent increases in domestic health spending, the country continues to rely on international donors for health systems financing [32]. Donors should be aware of potential unintended consequences to CHWs caused by aid stipulations and they should commit to strategies that support CHWs in their contributions to health systems strengthening.

## Acknowledgments

We extend our deepest appreciation to Community Health Workers and health center leaders who supported and participated in this work, and who dedicate their lives to improving health outcomes for all. We are also grateful to Mauwa Niyitegeka, Rachel Niyomufasha and Rosalie Uwayezu Nkunda for their assistance in collecting data and Dr. Zahirah McNatt for her recommendations on the qualitative analysis.

## Author Contributions

**Conceptualization:** Janna M. Schurer, Kelly Fowler, Ellen Rafferty, Ornella Masimbi, Olivia Rozanski, Hellen J. Amuguni.

**Data curation:** Janna M. Schurer.

**Formal analysis:** Janna M. Schurer, Kelly Fowler, Ellen Rafferty, Jean Muhire.

**Funding acquisition:** Hellen J. Amuguni.

**Investigation:** Janna M. Schurer, Kelly Fowler, Ellen Rafferty, Jean Muhire, Olivia Rozanski.

**Methodology:** Janna M. Schurer, Ellen Rafferty, Ornella Masimbi, Jean Muhire, Olivia Rozanski.

**Project administration:** Janna M. Schurer.

**Supervision:** Janna M. Schurer, Ellen Rafferty, Hellen J. Amuguni.

**Writing – original draft:** Janna M. Schurer.

**Writing – review & editing:** Kelly Fowler, Ellen Rafferty, Ornella Masimbi, Jean Muhire, Olivia Rozanski, Hellen J. Amuguni.

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
