## [Decision Letter · Decision Letter 0]

6 Jul 2020

Equity for health delivery: Opportunity costs and benefits among Community Health Workers in Rwanda

PONE-D-20-12651

Dear Dr. Schurer,

We’re pleased to inform you that your manuscript has been judged scientifically suitable for publication and will be formally accepted for publication once it meets all outstanding technical requirements.

Kind regards,

Joseph Telfair, DrPH, MSW, MPH

Academic Editor

PLOS ONE

Additional Editor Comments (optional):

Reviewers' comments:

Reviewer's Responses to Questions

**Comments to the Author**

1. Is the manuscript technically sound, and do the data support the conclusions?

Reviewer #1: Yes

Reviewer #2: Yes

2. Has the statistical analysis been performed appropriately and rigorously? 

Reviewer #1: Yes

Reviewer #2: Yes

3. Have the authors made all data underlying the findings in their manuscript fully available?

Reviewer #1: Yes

Reviewer #2: Yes

4. Is the manuscript presented in an intelligible fashion and written in standard English?

Reviewer #1: Yes

Reviewer #2: Yes

5. Review Comments to the Author

Reviewer #1: The work here presented adds additional insight to the knowledge about public health studies. The findings are important and related interests in this area of research. Would be good you can make full disclosure to the Rwanda Government to further improve the benefits among Community Health Workers.

Reviewer #2: This manuscript focuses on a very important issue: CHWs’ monetary compensation. The findings clearly indicate that CHW are not paid for their work nor supported with the necessary equipment to conduct their work well. The authors say this results in attrition. The quotes presented are eloquent. Below are my comments.

1. The authors stop short from saying explicitly that relying on poor people to do volunteer work to solve the most pressing health issues in a country is exploitation. Further, the good will and commitment of CHW is used to maintain the exploitative relationships. (Very few of us would give 18% of our own money to our employer.) The CHWs interviewed said that some of the problem for retention are family support. Reading about family support in the manuscript, it seems that some if it boils down to lack of financial compensation. A more in-depth discussion of this topic should be included.

2. Recommendations from the authors should be in the abstract.

3. A future study could interview those who dropped out and are no longer CHW to better understand the reasons for attrition. This could be discussed and is also a limitation, because only those who are still CHW were eligible for the study.

4. Methods:

a) The qualitative data comes from open-ended questions. It is not clear whether interviewers had the opportunity to follow up on responses, as they would have in a qualitative interview. That may be an additional limitation. (However, the data, as mentioned, is eloquent).

b)Was the 08 kappa for the entire transcript or for each code. Recently, literature recommends to calculate data per code.

c) On page 5 it says that they interviewed 10% of the national population. Is this the national population of CHW? Please clarify

5. Other:

a) There are some grammatical errors; the authors should check the manuscript.

6. PLOS authors have the option to publish the peer review history of their article (what does this mean?). If published, this will include your full peer review and any attached files.

Reviewer #1: No

Reviewer #2: **Yes: **Patricia I. Documet, MD, DrPH

---

## [Editor Report · Acceptance letter]

20 Jul 2020

PONE-D-20-12651 

Equity for health delivery: Opportunity costs and benefits among Community Health Workers in Rwanda 

Dear Dr. Schurer:

I'm pleased to inform you that your manuscript has been deemed suitable for publication in PLOS ONE. Congratulations! Your manuscript is now with our production department. 

Kind regards, 

on behalf of

Dr. Joseph Telfair 

Academic Editor

PLOS ONE